# Emerging Roles of Astrocyte Kir4.1 Channels in the Pathogenesis and Treatment of Brain Diseases

**DOI:** 10.3390/ijms221910236

**Published:** 2021-09-23

**Authors:** Yukihiro Ohno, Naofumi Kunisawa, Saki Shimizu

**Affiliations:** Department of Pharmacology, Faculty of Pharmacy, Osaka Medical and Pharmaceutical University, 4-20-1 Nasahara, Takatsuki, Osaka 569-1094, Japan; naofumi.kunisawa@ompu.ac.jp (N.K.); saki.shimizu@ompu.ac.jp (S.S.)

**Keywords:** astrocyte, Kir4.1 channel, spatial potassium buffering, mood disorder, epilepsy

## Abstract

Inwardly rectifying Kir4.1 channels in astrocytes mediate spatial potassium (K^+^) buffering, a clearance mechanism for excessive extracellular K^+^, in tripartite synapses. In addition to K^+^ homeostasis, astrocytic Kir4.1 channels also play an essential role in regulating extracellular glutamate levels via coupling with glutamate transporters. Moreover, Kir4.1 channels act as novel modulators of the expression of brain-derived neurotrophic factor (BDNF) in astrocytes. Specifically, inhibition of astrocytic Kir4.1 channels elevates extracellular K^+^ and glutamate levels at synapses and facilitates BDNF expression in astrocytes. These changes elevate neural excitability, which may facilitate synaptic plasticity and connectivity. In this article, we summarize the functions and pharmacological features of Kir4.1 channels in astrocytes and highlight the importance of these channels in the treatment of brain diseases. Although further validation in animal models and human patients is required, astrocytic Kir4.1 channel could potentially serve as a novel therapeutic target for the treatment of depressive disorders and epilepsy.

## 1. Introduction

In recent years, the classical concept of synapses has undergone revision, and the concept of tripartite synapses focusing on neuron–glia interactions has become stablished [1]. A single astrocyte connects to approximately 10^5^ synapses and wraps classical synaptic components (i.e., presynaptic nerve terminals and postsynaptic membranes of neural dendrites) with fine perisynaptic processes, forming tripartite synapses (Figure 1). Perisynaptic astrocytes maintain ion and water homeostasis, take up and metabolize neurotransmitters, and secrete a range of neuroactive substances, including gliotransmitters (e.g., glutamate, d-serine, and ATP) and neurotrophic factors (e.g., glial cell-line derived neurotrophic factor (GDNF) and brain-derived neurotrophic factor (BDNF)) [1]. Astrocytes are tightly interconnected by gap junctions, and their perivascular foot processes form blood brain barriers with microcapillaries (Figure 1). Thus, astrocytes are essential for maintaining normal brain functions and are thought to be closely involved in the pathogenesis and treatment of brain diseases.

Among the diverse functions of astrocytes, spatial potassium (K^+^) buffering is of particular importance with respect to the regulation of neural excitability (Figure 1) [2,3]. This buffering process serves as a clearance system for excessive extracellular K^+^ in synapses and is primarily mediated by inwardly rectifying potassium (Kir) channels containing Kir4.1 subunits (Kir4.1 channels) [4,5]. Kir4.1 channel-mediated spatial K^+^ buffering is also important for glutamate and water homeostasis in synapses, as Kir4.1 channels are functionally coupled to the excitatory amino acid transporters (EAATs), EAAT1 (glutamate aspartate transporter 1: GLAST-1) and EAAT2 (glutamate transporter 2: GLT1), and the water transporter aquaporin-4 (AQP4) [2,4,5,6]. Moreover, it has recently been shown that these channels modulate the expression of BDNF in astrocytes [7]. It is therefore probable that astrocyte Kir4.1 channels regulate both the excitability and synaptic plasticity of neurons by controlling extracellular levels of K^+^ ions, glutamate, and BDNF in tripartite synapses, and are consequently involved in the development of different brain diseases. Here, we summarize the functions and pharmacological features of Kir4.1 channels in astrocytes and discuss their therapeutic potential in modulating brain diseases, specifically epileptic and mood disorders.

## 2. Astrocytic Spatial K^+^ Buffering and Kir4.1 Channels 

Ion homeostasis in tripartite synapses, especially the control of extracellular K^+^ concentration ([K^+^]_o_) that determines the resting membrane potential of neurons, is essential for maintaining normal brain activities [2,3]. Neural excitation elicits an influx of Na^+^, and in turn a K^+^ efflux, thereby promoting the local elevation of [K^+^]_o_ during the repolarization process. The spatial K^+^ buffering controlled by astrocytes normalizes elevated extracellular K^+^ levels and maintains [K^+^]_o_ at approximately concentrations of 3–5 mM (Figure 1). Even during seizure discharges, the accumulation of extracellular K^+^ shows a ceiling level of 10–12 mM [K^+^]_o_, which is maintained by the K^+^ buffering function of astrocytes [2,3,8]. However, in response to disruption of the K^+^ buffering, [K^+^]_o_ can increase to levels 25–80 mM which causes spreading depression waves [3,8]. Thus, spatial K^+^ buffering by astrocytes acts as one of major clearance systems for excessive extracellular K^+^ at synapses. Given that astrocytes normally have a more hyperpolarized resting membrane potential than neurons, locally elevated extracellular K^+^ readily flows into astrocytes than neurons through Kir4.1 channels located on the membranes of perisynaptic processes. The inflowing K^+^ ions subsequently diffuse passively throughout astrocytic syncytium and are transported to sites of low [K^+^]_o_, such as microcapillaries (Figure 1). Computational studies of an astrocyte–neuron network model have revealed that repeated neural firing produces a substantial increase in [K^+^]_o_ from 3 mM to almost 10 mM, which can then be rapidly normalized by perisynaptic astrocyte Kir4.1 channels [8]. In addition, when Kir4.1 channel conductance (*G_Kir_*) is suppressed, K^+^ uptake by astrocytes is markedly inhibited and the peak level of [K^+^]_o_ in synapses is elevated to 15 mM or more. The elevated [K^+^]_o_ depolarizes the membrane potential of the surrounding neurons, thereby generating seizure-like burst firing [8]. In this regard, it has been estimated that the critical value of *G_Kir_* for the generation of spontaneous epileptic seizures in the absence of external stimuli is approximately 7 *pS* or less. Furthermore, Kir4.1 channels are known to be essential in maintaining the resting membrane potential of astrocytes, and inhibition of Kir4.1 channel activity has been shown to cause astrocyte depolarization [8,9,10,11].

Within astrocytes, Kir4.1 channel-mediated spatial K^+^ buffering is functionally linked to EAATs and plays an important role in glutamate homeostasis in tripartite synapses [2,6,10,11]. Kir4.1 channels in the membranes of perisynaptic astrocyte processes colocalize with EAAT1 and EAAT2, as well as the water transporter AQP4 (Figure 1) [4,6,10,12]. Specifically, EAAT2 is the major EAAT isoform, which is expressed predominantly in astrocytes and is responsible for approximately 90% of the total synaptic glutamate clearance [13]. It co-transports glutamate with three Na^+^ ions and a single H^+^ ion and counter-transports with one K^+^ ion (Figure 2). Accordingly, its kinetics are highly Na^+^-, K^+^-, and voltage-dependent [6,14,15]. Thus, glutamate uptake by EAAT2 into astrocytes is regulated by an electrochemical gradient of Na^+^/K^+^ ions and the resting membrane potential of astrocytes. Given that Kir4.1 channel-mediated K^+^ buffering is important for the maintenance of strong hyperpolarization of the resting membrane potential of astrocytes [9,10,11], the activity of these channels has a pronounced effect on the glutamate uptake capacity of EAAT2 in astrocytes. Indeed, inhibition of Kir4.1 channels causes astrocyte membrane depolarization and suppresses glutamate uptake by reducing the driving force for glutamate transport [6,11,12]. It is therefore probable that Kir4.1 channel-mediated spatial K^+^ buffering is essential not only for K^+^ homeostasis but also for regulating the extracellular concentration of glutamate ([glutamate]_o_) in tripartite synapses.

## 3. Molecular Pharmacology of Kir4.1 Channels

The *KCNJ10* (human) or *Kcnj10* (rodents) gene, which encodes the Kir4.1 subunit, is located on chromosome (Chr) 1 (1q23.2) in humans, Chr 1 (1H3) in mice and Chr 13 (13q24) in rats, encoding 379, 476, and 381 amino acids, respectively [4,5]. The primary structure of the Kir4.1 subunit consists of two transmembrane (TM) regions, an extracellular pore-forming region, which has a -G-Y-G- signature sequence acting as an ion filter, and intracellular N- and C-terminal domains (Figure 1). The Kir4.1 subunit constructs Kir4.1 channels by assembling homotetramers, which have a weak inwardly rectifying property, conducting a large inward and moderate outward K^+^ current. The Kir4.1 subunit also constructs Kir4.1/5.1 channels by forming hetero tetramers with a further Kir subunit, Kir5.1. The *G_Kir_* of Kir4.1 channels ranges between 20 and 40 *pS* and that of Kir4.1/5.1 channels between 40 and 60 *pS* [4]. Importantly, Kir4.1 subunits play essential roles in maintaining the activity of Kir4.1/5.1 channels, as evidence by the finding that loss-of-function mutations of Kir4.1 subunits suppress the activity of Kir4.1/5.1 channels constructed with normal Kir 5.1 subunits [16,17]. 

Although for long, the agents that act on Kir4.1 channels remained undetermined, several compounds that specifically interact with Kir4.1 have now been identified (Table 1). For example, among the drugs that affect the central nervous system, we have demonstrated, for the first time, that several antidepressants block Kir4.1 channels heterologously expressed in HEK293 cells [18,19]. Whole-cell patch-clamp analyses have revealed that tricyclic antidepressants (TCAs; e.g., nortriptyline, desipramine and imipramine) and selective serotonin reuptake inhibitors (SSRIs; e.g., fluoxetine and sertraline) block Kir4.1 channel currents in a reversible concentration- and subunit-dependent manner, the former inhibition being voltage-dependent but the later voltage-independent. Moreover, the blocking effects of antidepressants appear to be independent on the actions of these drugs on the serotonergic nervous system (i.e., inhibition of 5-HT reuptake or stimulation of 5-HT receptors), and studies using Kir4.1/Kir1.1-chimeric channels have shown that antidepressants specifically interact with the TM region of Kir4.1 subunits to block the construction of Kir4.1 channels [20]. Furthermore, on the basis of alanine-scanning mutagenesis analyses, two amino acid residues, T128 and E158, have been identified as primary sites associated with the action of effective antidepressants (Figure 1). T128 is located in the inner pore region immediately below the G-Y-G sequence, whereas E158 lies within the center of the TM2 region. Combined pharmacophore and molecular docking analyses have revealed that T128 can bind to the benzene ring (hydrogen bond acceptor) of antidepressants via a hydrogen bond, whereas E158 bind to the amine moiety of these rings via an ionic bond [20]. Since the IC_50_ value of fluoxetine (15.2 μM) is similar to its brain concentrations (ca. 17–25 μM) in the chronic treatments for 3–5 weeks at a clinical setting [19], the inhibition of astrocytic Kir4.1 channels may be partly involved in the clinical actions of antidepressant drugs. 

More recently, studies by other groups have revealed that Kir4.1 channels are also blocked by several additional compounds, including quinacrine [21], chloroquine [22], pentamidine [23], VU0134992 [24], and aminoglycoside antibiotics [25] (Table 1). The antimalarial drugs quinacrine and chloroquine (3–30 μM) inhibit Kir4.1 channels in a voltage-dependent manner in a whole-cell patch configuration, and similarly, pentamidine (0.03–3 μM), an orphan drug for *Pneumocyctis carinii*, has been shown to induce voltage-dependent inhibition of Kir4.1 channels. Interestingly, these agents show more potent blocking actions (IC_50_ = 0.1–2 μM) in inside-out patches. Although the potencies are relatively mild, ototoxic aminoglycoside antibiotics (e.g., gentamicin and neomycin, 3–1000 μM), have also been identified as voltage-dependent blockers of Kir4.1 channels. Importantly in this respect, molecular modeling and site-directed mutagenesis studies have shown that in a manner similar to that of antidepressant drugs (SSRIs and TCAs), all these Kir4.1 inhibitors bind primarily to E158 in the TM2 region and T128 adjacent to the G-Y-G motif to block Kir4.1 channels (Figure 1) [21,22,23,24,25]. Although the clinical relevance of the Kir4.1 channels by above agents remains unknown, Kir4.1 block by aminoglycoside antibiotics are suggested to be involved in their ototoxic effects [25]. 

Thus, pentamidine (IC_50_ = 0.097 μM) is the most potent inhibitor and VU0134992 is the second (IC_50_ = 0.97 μM) among the compounds reported. There is no specific inhibitor for Kir4.1 channels so far, nonetheless, the information obtained from site-directed mutagenesis and Kir4.1 molecular modeling studies will make an important contribution to designing novel specific ligands for Kir4.1 channels in the future.

As previously described, inhibition of Kir4.1 channels impairs the spatial K^+^ buffering function of astrocytes and elevates both [K^+^]_o_ and [glutamate]_o_ in tripartite synapses, thereby inducing neural excitation (Figure 2). In addition, it has recently been demonstrated that blockade of Kir4.1 channels by antidepressants markedly enhances the expression of BDNF in astrocytes [7]. Although antidepressants commonly inhibit 5-HT reuptake and fluoxetine is known to enhance BDNF expression by blocking GluN2B-containing NMDA receptors [26], the relative potencies of antidepressants for Kir4.1 channel inhibition (sertraline > fluoxetine > imipramine >> fluvoxamine > mianserin) are consistent with those for BDNF induction. Moreover, suppression of Kir4.1 expression by siRNA transfection has been observed to markedly enhance BDNF expression that had been specifically antagonized using an ERK inhibitor [7]. Furthermore, down-regulation of Kir4.1 expression has also been shown to significantly elevate the expression of GDNF and the ciliary neurotrophic factor, although their responses were considerably weaker than those of BDNF. Although these observations were obtained from in vitro culture studies so far, it is probable that Kir4.1 channels regulate not only neural excitability via K^+^ and glutamate transport, but also astrocytic BDNF expression via the Ras/ERK/CREB signaling pathway (Figure 2). Given that BDNF is a key molecule that affects a range of brain functions, including neuronal and glial development, neuroprotection, synaptic plasticity, and different pathophysiological responses to brain disorders [5,27,28,29], the Kir4.1-BDNF system is assumed to play important roles in modulating and modifying the brain diseases.

## 4. Kir4.1 Channels and Brain Diseases

### 4.1. Major Depressive Disorder (MDD)

Depressive disorders (e.g., MDD, depressive mood and dysthymia) are the most common mental disorders, affecting approximately 5–20% of the population worldwide [30]. Patients with MDD show diverse symptoms including anhedonia (inability to experience pleasure), blunted mood, emotional withdrawal, reduced activity, cognitive impairment, sleep disturbance and psychosomatic symptoms. As the neurobiological basis, MDD is associated with the depletion of monoamines (particularly 5-HT and norepinephrine) in the brain (monoamine hypothesis). All current antidepressants have been developed solely based on the “monoamine hypothesis,” with most drugs inhibiting the reuptake of 5-HT and/or norepinephrine by monoamine transporters into nerve terminals. SSRIs are the current first-line antidepressant drugs; however, there are significant deficiencies in the current treatment of MDD that need to be overcome, including (1) the slow onset of antidepressant effects, (2) the presence of treatment-refractory symptoms, and (3) various adverse reactions such as gastrointestinal side effects, serotonin syndrome, and exacerbation of neuroleptic malignant syndrome [31,32]. Therefore, the development of new drugs that can be used to rectify these deficiencies is considered a priority. 

In addition to the “monoamine hypothesis”, it is well established that MDD is associated with morphological atrophy of specific brain structures such as the medial prefrontal cortex and limbic areas (e.g., amygdala and hippocampus) [31,33,34,35]. Brain atrophy in MDD is attributed to a reduction in the number of neurons (both glutamatergic and GABAergic neurons) and astrocytes, accompanied by deteriorative changes in dendritic spines and synaptic connectivity. These changes may be caused, at least partially, by the stress-induced release of adrenal glucocorticoids and microglial inflammatory cytokines. In addition, degenerative changes in the brains of patients with MDD are believed to be a consequence of the reduced expression of BDNF in response to exposure to chronic stress [28,31,33,34,35]. Indeed, multiple missense polymorphisms of the BDNF gene have been reported to be associated with MDD [36]. Moreover, antidepressant drugs have been observed to elevate the expression of BDNF in both neurons and astrocytes and restore impaired neurogenesis in the hippocampus and prefrontal cortex [28,37,38].

As described previously, multiple types of antidepressant drug inhibit Kir4.1 channels and elevate astrocytic BDNF expression, suggesting that astrocytic Kir4.1 channels can serve as a therapeutic target for the treatment of MDD [5,6]. Kir4.1 channels in astrocytes can modulate the development of MDD via dual mechanisms, though altering the K^+^ buffering function and expression of BDNF (Figure 2). Inhibition of Kir4.1 channels attenuates the K^+^ buffering and elevates the levels of [K^+^]_o_ and [glutamate]_o_, thereby increasing neural excitability in synapses. It also facilitates astrocytic BDNF expression, which may in turn alleviate degenerative alterations in the brain of MDD patients (Figure 3). Conversely, activation of Kir4.1 channels (expressional up-regulation or channel stimulation) presumably facilitates the development of MDD by lowering the synaptic levels of [K^+^]_o_ and [glutamate]_o_, and by reducing the expression of BDNF in astrocytes. Consistent with the potential role of astrocytic Kir4.1 channels in the pathogenesis and treatment of MDD, recent studies have shown that astrocytic Kir4.1 channels in the lateral habenula (LHb) are involved in altering the firing mode of LHb neurons and modulating the induction of depressive behaviors in an animal model of depression [39]. These authors demonstrated that gain-of-function transgenic treatment of Kir4.1 channels is associated with a hyperpolarization of both neurons and astrocytes in the LHb, which has the effect of producing burst firings of LHb neurons and depression-like behaviors in congenitally learned helpless rats. In addition, loss-of-function (expressional knockdown) of Kir4.1 channels in the LHb induces a depolarization of neurons, shifting the burst firings to tonic firing patterns, and ameliorating depression-like behaviors. Although information relating to pathophysiological alterations in Kir4.1 channels of human patients is still limited, the findings of a recent study [40] have indicated that the level of Kir4.1 expression was significantly higher in the parietal cortex of MDD patients than in a control group, thereby providing evidence in support of the pathogenic role of Kir4.1 channels in MDD.

Among the drawbacks of the current treatment of MDD, a significant time lag in the efficacy onset and the presence of treatment-refractory symptoms are of particular note. To overcome these clinical restrictions, ketamine has been gaining worldwide attention as a rapid-acting antidepressant agent [6,35,41,42,43]. Specifically, sub-anesthetic doses of ketamine have been found to produce rapid and persistent antidepressant effects in patients with MDD, even in treatment-resistant cases [42]. Although the precise mechanisms underlying the action of ketamine are currently under investigation, emerging evidence indicates that the glutamate–BDNF system is closely associated with the rapid onset of its antidepressant effects (Figure 3). Ketamine elevates [glutamate]_o_ in the prefrontal cortex and hippocampus, which in turn activates the BDNF–TrkB pathway and augments synaptic connectivity by activating the mammalian target of rapamycin complex 1 (mTORC1) signaling [35,44]. These findings strongly suggest that antidepressants can alleviate degenerative changes in the brain of depressive patients by activating the glutamate-BDNF system, which is presumably linked to the mechanism of action underlying the rapid-acting antidepressant effect of ketamine (Figure 3). Furthermore, astrocytic Kir4.1 channels have been suggested to be involved in the rapid onset of the action of ketamine [6,43,45,46]. Indeed, ketamine has been found to suppress the burst firings of LHb neurons, which are regulated by Kir4.1 in astrocytes [39], and rapidly alleviates depressive behaviors in depression model rats [45]. Moreover, ketamine has been demonstrated to reduce Kir4.1 channel expression in astrocytic plasma membranes by inhibiting the trafficking of cytoplasmic Kir4.1 to membranes, which consequently suppresses Kir4.1 channel activity in astrocytes [47]. Thus, it can be conjectured that astrocytic Kir4.1 channels in the LHb are implicated in the rapid-acting antidepressant action of ketamine [6,44,46]. Collectively, the findings of aforementioned studies provide persuasive evidence that Kir4.1 channel blockers or expressional suppressors of Kir4.1 channels are attractive candidates as novel therapeutic agents for the treatment of depressive disorders (Figure 4).

### 4.2. Epileptic Disorders

Epilepsy is a chronic neurological disorder accompanied by recurrent epileptic seizures that affects approximately 50 to 70 million people worldwide [48]. Various antiepileptic agents that act primarily on neurons have been used in the treatment of epilepsy, including blockers of voltage-gated Na^+^ and Ca^2+^ channels, GABA_A_ receptor stimulants, and glutamate receptor antagonists [49]. Nonetheless, nearly one-third of patients are refractory to treatment with current medications. In addition, all anti-epileptics are neural depressants, and their actions on the development of chronic epilepsy (epileptogenesis) have yet to be clarified. Moreover, the use of these agents is often associated with serious side effects, including oversedation, arrhythmia, and serious mucocutaneous adverse reactions, such as Stevens–Johnson syndrome, as well as toxic epidermal necrolysis and teratogenesis [5,49]. Thus, new medications with innovative mechanisms of action are needed for the treatment of epilepsy, and in this regard, astrocytic Kir4.1 channels may provide a promising therapeutic target for countering epileptogenesis.

Studies using conditional knockout animals targeting astrocytic Kir4.1 have revealed a clear link between Kir4.1 channel dysfunction and seizure susceptibility (Table 2) [9,11]. These animals exhibited marked body tremors, ataxia, and high susceptibility to generalized tonic-clonic seizures (GTCS). Genetic deletion of Kir4.1 suppresses the uptake of both K^+^ and glutamate into astrocytes, thereby indicating that dysfunction of Kir4.1 channels elevate the susceptibility to GTCS by elevating extracellular levels of K^+^ and glutamate (Figure 2) [11,12]. These events are supported by computational modeling studies of tripartite synapses [8], which have shown that a reduction in Kir4.1 channel conductance (*G_Kir_*: 45 to 5 *pS*) markedly increases [K^+^]_o_ levels and evokes concomitant neural depolarization, thus producing spontaneous periodic seizure discharges. Importantly, Kir4.1 knockout has been demonstrated to promote a depolarization of membrane potential in astrocytes [8,9,12], which has the effect of hampering the voltage-dependent glutamate uptake into astrocytes by EAATs. These findings thus indicate that the suppression of K^+^ and glutamate uptake into astrocytes may be associated with seizure generation (ictogenesis) following Kir4.1 dysfunction.

Pathophysiological alterations of the Kir4.1 channels have been reported in various animal models of epilepsy (Table 2). For example, seizure-susceptible DBA/2 mice carry a T262A variation in the *Kcnj10* gene and exhibit reduced capacity for K^+^ and glutamate uptake into astrocytes [50]. Furthermore, a rat model of human autosomal dominant lateral temporal lobe epilepsy (ADLTE), carrying a mutation (L385R) in the *leucine-richglioma-inactivated 1* (*Lgi1*) gene, shows reduced expression of astrocytic Kir4.1 channels in the cerebral cortex and limbic regions (i.e., hippocampus and amygdala) [51], whereas Noda epileptic rats (NER), a hereditary epilepsy model exhibiting spontaneous GTCS, are characterized by a region-specific reduction in Kir4.1 of the amygdala, specifically in the astrocyte perisynaptic processes surrounding neurons [52]. Other models of epileptic disorders (e.g., a brain injury-induced seizure model [53] and an extravasated albumin-induced seizure model [54]) are also characterized by downregulation of Kir4.1 expression in the neocortex and hippocampus. Since several models (e.g., NER and trauma-induced show selective loss of Kir4.1 in astrocytic processes without changes in their somata [51,52], it is suggested that certain disease conditions affect the trafficking (subcellular distribution) process of Kir4.1. Collectively, the findings of these studies provide convincing evidence in support of the view that the impaired functioning of astrocytic Kir4.1 channels is closely linked to seizure generation and epileptogenesis (Figure 2). 

The causal role of Kir4.1 channels in epileptogenesis have been confirmed in patients with epileptic disorders (Table 2). Loss-of-function mutations in the *KCNJ10* gene have been found to elicit “EAST” (epilepsy, ataxia, sensorineural deafness, and tubulopathy) [55] or “SeSAME” (seizures, sensorineural deafness, ataxia, mental retardation, and electrolyte imbalance) [56] syndrome. Frequent mutations of the *KCNJ10* gene in EAST/SeSAME patients include R65P and G77R in the TM1 region, C140R in an extracellular loop between TM1 and TM2, T164I and A167V in the TM2 region, and R175Q, R199X and R297C in the C-terminal region [16,55,56,57]. All of these mutations are associated with marked inhibition of Kir4.1 channels (not only Kir4.1 homotetramers, but also Kir4.1/5.1 hetero-tetramers), thereby causing an elevation of extracellular K^+^ and glutamate levels. Furthermore, it has been reported that the number of Kir4.1 channels are reduced in hippocampal specimens obtained from patients with refractory temporal lobe epilepsy (TLE) [58]. Astrocytic Kir4.1 expression has been shown to be reduced in TLE patients with hippocampal sclerosis [58,59], as well as those patients with refractory partial epilepsy diagnosed as “focal cortical dysplasia type 1” [60]. Dysfunctions of Kir4.1 channels are also reported in patients with astrocytic tumor [61] or lethal severe-disabling seizures [62]. On the basis of the aforementioned observations, it would thus appear probable that Kir4.1 channel dysfunction is implicated in the pathogenesis of human epileptic disorders.

Given that inhibition (i.e., channel blockade and translational knockdown) of Kir4.1 channels has the effect of inducing the astrocytic expression of BDNF, and a BDNF is an established epileptogenic factor [5,7], it would appear that Kir4.1 channels may in part modulate the development of chronic epilepsy by elevating astrocytic BDNF expression (Figure 2). BDNF enhances neural sprouting, synaptic plasticity, and astrogliosis, thereby facilitating epileptogenesis [28], and elevated expression of BDNF and TrkB (BDNF receptor) has been reported in different animal models of epilepsy and in human patients with epilepsy [27]. In addition, it has been demonstrated that kindling development, a model of epileptogenesis, in animals is suppressed by expressional knockdown of BDNF and inhibition of TrkB signaling [27]. Thus, the Kir4.1-BDNF system in astrocytes may play a role in the pathogenesis of some epileptic disorders.

Although to date there have been no cures that can specifically ameliorate epileptogenic processes, “channel activators” or “expressional enhancers” of Kir4.1 channels may become the novel medication for epileptogenesis (Figure 4). It is anticipated that these agents would contribute to alleviating epilepsy by enhancing the K^+^ buffering function of astrocytes, thereby lowering both [K^+^]_o_ and [glutamate]_o_ levels in tripartite synapses, and by reducing the astrocyte expression and secretion of BDNF. Interestingly, we have shown that repeated treatment of animals with several antiepileptic drugs (e.g., valproate, phenytoin, and phenobarbital), which are effective for GTCS in humans, elevated Kir4.1 expression in astrocytes without altering astrocytes numbers [63]. In addition, prophylactic treatment of animals using valproate has been found to prevent both the reduced expression of astrocytic Kir4.1 channels and seizure sensitization during audiogenic epileptogenesis (kindling) in *Lgi1*-mutant rats [51]. These actions may be related to the prophylactic effects of antiepileptics in the chronic treatment of epilepsy [64], thereby providing support for the therapeutic potential of “expressional enhancers” of astrocytic Kir4.1 channels against epileptogenesis. Other agents known to elevate the expression of Kir4.1 channels include guanosine [65], dexamethasone [66], and minocycline [67]; however, their effects in epilepsy models have yet to be investigated. Further proof of concept studies are necessary, and the mechanisms underlying the expressional enhancement of Kir4.1 channels remain to be clarified. 

### 4.3. Other Brain Diseases

It has also been suggested that Kir4.1 channels are involved in a number of brain diseases, including Huntington’s disease, autism spectrum disorders, Parkinson’s disease, Alzheimer’s disease, amyotrophic lateral sclerosis, and neuropathic pain.

Patients with Huntington’s disease manifest progressive extrapyramidal movement disorders, cognitive impairment, and psychiatric disturbance [68,69,70]. The disease is an inherited neurodegenerative disorder associated with an autosomal dominant mutation in the Huntingtin gene (*HTT*). Mutant *HTT* encodes a protein containing an expanded chain of polyglutamines in the N-terminal region, which promotes the intracellular aggregation of HTT and causes neural damage in the basal ganglia (e.g., striatum and globus pallidus), cortical regions, and hippocampus [68]. It has been established that mutant HTT accumulates and damages neurons as well as astrocytes in the striatum [69,70,71,72,73]. In addition, Kir4.1 channels and EAATs (particularly EAAT2) in astrocytes have been shown to be reduced in patients with Huntington’s disease and animal models. Given that reductions in the activity of Kir4.1 channels and EAATs result in an elevation of [K^+^]_o_ and [glutamate]_o_, these changes also have excitotoxic effects on neurons in the basal ganglia (e.g., GABAergic medium spiny neurons in the striatum). Thus, it may be possible that enhancement of Kir4.1 channel expression or activation of Kir4.1 channels can reduce the damage in striatal neurons and prevent the development of Huntington’s disease (Figure 4). A comprehensive review of the roles of astrocytic Kir4.1 channels in the pathogenesis of Huntington’s disease has been published elsewhere [69]. 

Autism spectrum disorder is a developmental disease that affects the communication and behavior of affected individuals (e.g., difficulties with social communication and interaction, restricted interests, stereotypical behaviors, and affective disorders), which is characterized by a wide variation in the types and severity of symptoms. In addition, it is known that autism spectrum disorder and epilepsy frequently co-occur: epilepsy occurs in 6–27% of patients with autism spectrum disorder, while the rate of autism diagnosis is estimated from 5% to 37% in epileptic children [74]. Interestingly, gain-of-function defects (i.e., missense mutations of R18Q at the N terminus and of V84M in the TM1 region) in astrocytic Kir4.1 channels have been detected in children with autism spectrum disorders and epilepsy (Figure 4) [75,76], although information regarding the association of Kir4.1 channels with these disorders is still limited. Moreover, expressional down-regulation of Kir4.1 channels in astrocytes has also been reported in animal models of amyotrophic lateral sclerosis [77,78], although the pathophysiological mechanisms of Kir4.1 channels underlying the development of these diseases have yet to be ascertained.

Neuropathic pain is a chronic disease caused by neural damage or as a consequence of diseases affecting the somatosensory nervous system, which is often accompanied by abnormal sensations (e.g., dysesthesia and allodynia) [79]. Several studies have shown that Kir4.1 channels in the satellite glial cells of sensory ganglia (dorsal root ganglia and trigeminal ganglia) are down-regulated in animal models of neuropathic pain [80]. In addition, knockdown of Kir4.1 using RNA interference techniques in the trigeminal ganglion has been observed to evoke facial pain-like behaviors [81]. Given that BDNF is a key factor in the etiology of neuropathic pain [82], it can be speculated that the Kir4.1–BDNF system in glial cells is also involved in the chronic pain sensitization associated with neuropathic pain (Figure 4). Details regarding the function of Kir4.1 channels in satellite glial cells in modulating chronic pain have been described elsewhere [80].

## 5. Concluding Remarks

Astrocytic Kir4.1 channels play an essential role in regulating the extracellular levels of K^+^ and glutamate, and the astrocytic expression of BDNF in tripartite synapses. Accumulating evidence indicates that inhibition of Kir4.1 channels attenuates spatial K^+^ buffering by astrocytes, increases the levels of [K^+^]_o_ and [glutamate]_o_, excites surrounding neurons, and enhances the astrocytic expression of BDNF. These events in turn appear to evoke seizures (ictogenesis) and contribute to the development of chronic epilepsy (epileptogenesis). In contrast, overactivation of Kir4.1 channels diminishes neuronal excitability and attenuates astrocytic BDNF expression, which appears to be associated with the development of MDD. We highlight the therapeutic potential of Kir4.1 channel modulators in brain diseases, specifically Kir4.1 channel stimulators (e.g., channel activators and expressional enhancers) for epileptic disorders and other diseases such as Huntington’s disease and neuropathic pain, and Kir4.1 channel inhibitors (e.g., channel blockers and expressional suppressors) for depressive disorders and autism (Figure 4). These agents are attractive candidates as the novel medications that would act primarily on glial cells in the treatment of brain diseases. Further investigations, particularly on the control of Kir4.1 expression, are important for drug discovery researches that seek to identify novel Kir4.1 channel modulators (expression enhancers or suppressors). While a recent study suggests that dietary K^+^ intake plays an important role in regulation of Kir4.1 expression in basolateral membranes of the renal distal convoluted tubule [83], influences of dietary K^+^ on Kir4.1 expression in the brain remain unknown. Nonetheless, several agents (e.g., guanosine and dexamethasone) have been shown to enhance the expression of Kir4.1 channels [66,67] and DNA methylation is suggested to be involved in regulation of Kir4.1 transcriptional activity [84]. Further studies are required to clarify the molecular mechanisms underlying the epigenetic regulation of Kir4.1. In addition, while the functions of astrocytic Kir4.1 channels in regulating K^+^ and glutamate homeostasis at tripartite synapses are well supported by numerous studies, their functions and mechanisms in regulating BDNF expression remain to be elucidated. Particularly, in vivo studies using animal models and human patients are necessary to delineate the detailed function of the Kir4.1-BDNF system in modulating the pathophysiology of brain disease.

## Figures and Tables

**Figure 1 ijms-22-10236-f001:**
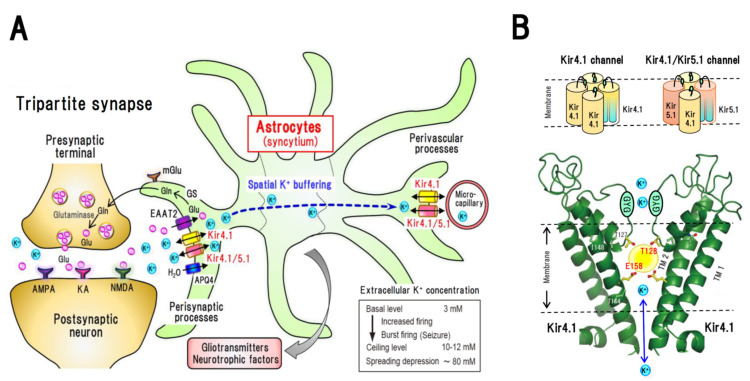
Spatial potassium (K^+^) buffering by astrocytes in tripartite synapses and inwardly rectifying Kir4.1 channels. (**A**): Kir4.1 channel-mediated spatial K^+^ buffering function of astrocytes. The Kir channel subunits Kir4.1 and Kir5.1 are specifically expressed in astrocytes and co-localized with excitatory amino acid transporter 2 (EAAT2) and aquaporin 4 (APQ4) in the perisynaptic processes. Astrocytic Kir4.1 channels mediate spatial K^+^ buffering, a clearance system for excessive extracellular K^+^ in synapses, controlling the extracellular K^+^ levels at concentrations of 3–5 mM and maintaining the ceiling level of around 10–12 mM even during seizure discharges. When the function of Kir4.1 channels is disrupted under disease conditions, extracellular K^+^ levels can readily increase to induce abnormal firing of neurons. AMPA: α-amino-3-hydroxy-5-methyl-4-isoxazolepropionic acid (AMPA) receptor, GS: glutamine synthetase, KA: kainic acid (KA) receptor, NMDA: *N*-methyl-d-aspartate (NMDA) receptor, mGlu: metabotropic glutamate receptor. (**B**): Structure of Kir4.1 channels. Kir4.1 subunits construct Kir4.1 channels by assembling homo-tetramers and Kir4.1/5.1 channels by forming hetero-tetramers with Kir5.1 subunit. Kir4.1 subunits possess two transmembrane (TM) regions, TM1 and TM2, between which an ion-selective filter containing a -G-Y-G- signature motif is located. Several compound including antidepressant drugs (e.g., fluoxetine) bind to and inhibit Kir4.1 channel-mediated currents. Two amino acid residues, T128 and E158, within the Kir4.1 subunit have been identified as specific binding sites for antidepressants and other blockers, including quinacrine, chloroquine, pentamidine, VU0134992 and aminoglycoside antibiotics. A yellow circle represents a drug-binding pocket (approximately 200 Å^3^) estimated by the docking simulation analysis for antidepressants, where the benzene ring of antidepressants binds to T128 via a hydrogen bond and the amine moiety of these rings bind to E158 via an ionic bond.

**Figure 2 ijms-22-10236-f002:**
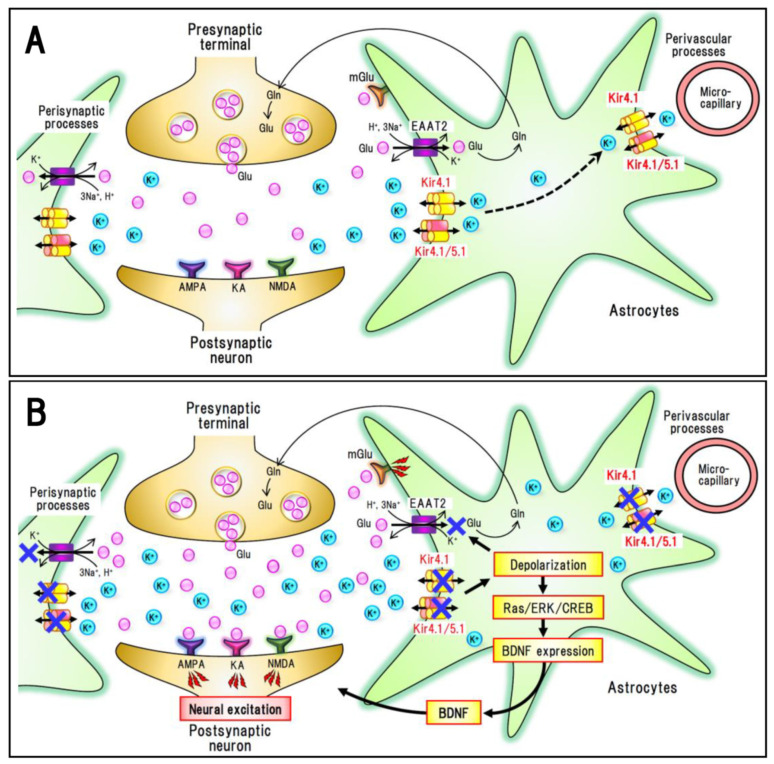
Diagrams illustrating the pathophysiological changes in Kir4.1 channel functions. (**A**) Synapses under normal conditions. (**B**) Synapses in disease conditions. The functions of astrocytic Kir4.1 channels are inhibited, under pathophysiological conditions, by gene mutations, expressional suppression, or pharmacological treatments. Inhibition of Kir4.1 channels results in an elevation of extracellular K^+^ levels in synapses. Since Kir4.1 channels play a role in maintaining the resting membrane potential of astrocytes, Kir4.1 channel blockade causes a depolarization (excitation) of astrocytes, which in turn inhibits astrocytic glutamate uptake via EAAT2. Elevated levels of extracellular K^+^ and glutamate cause a hyperexcitation of neurons. Astrocyte excitation also promotes an increase in the expression of BDNF by activating the Res/ERK/CREB pathway, which may consequently contribute to epileptogenesis via enhancing neural connectivity, synaptic plasticity, and neurogenesis.

**Figure 3 ijms-22-10236-f003:**
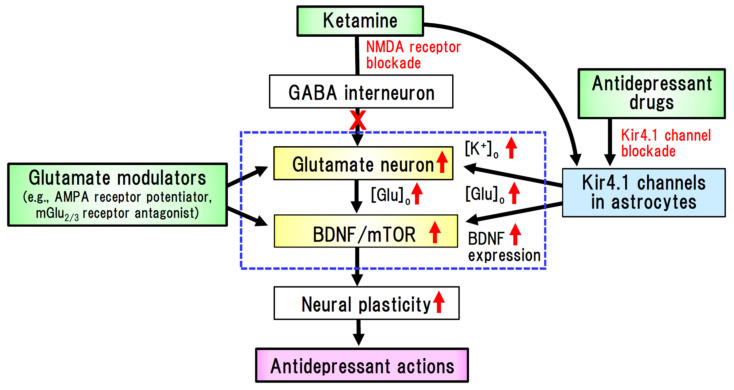
New medications for the treatment of major depressive disorders (MDD) based on the glutamate–BDNF hypothesis. Ketamine is known as a rapid-acting antidepressant drug with clinically proved efficacy, the activity of which is believed to be associated with the glutamate–BDNF–mTOR system. Ketamine inhibits NMDA receptors in GABA interneurons, which in turn activates glutamatergic neurons and elevates extracellular levels of glutamate. Glutamatergic neuron activation facilitates neural plasticity and alleviates morphological changes in the brain of MDD patients. This drug also induces the expression of BDNF in astrocytes via its interaction with Kir4.1 channels or a non-NMDA mechanism (e.g., Gs-cAMP signaling). Kir4.1 channel inhibitors presumably have mechanisms of action similar to those of ketamine, promoting the elevation of extracellular glutamate and inducing BDNF expression. Glutamate modulators, such as AMPA receptor potentiators and presynaptic mGlu_2/3_ receptor antagonists, similarly mimic the action of ketamine via activating glutamatergic neurons.

**Figure 4 ijms-22-10236-f004:**
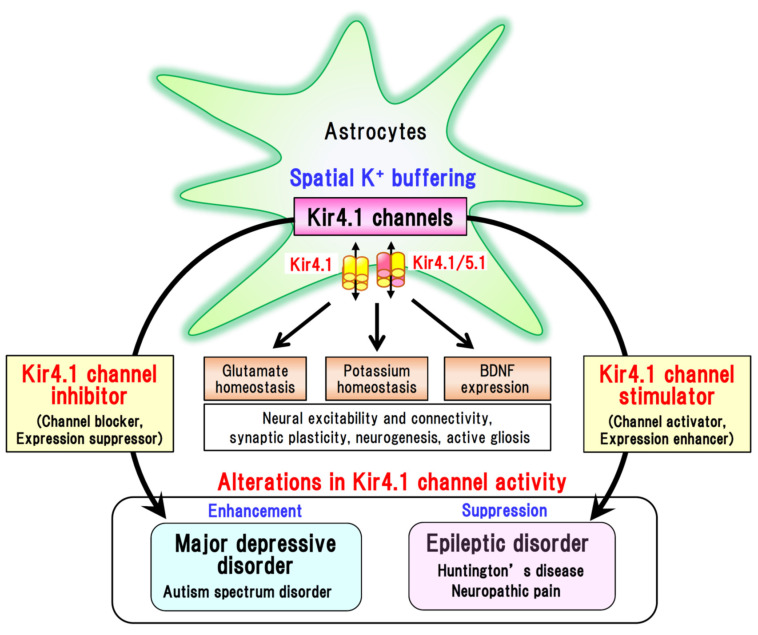
Dysfunction of Kir4.1 channels in brain diseases and potential glia-oriented therapeutic strategies. Kir4.1 channels mediate spatial K^+^ buffering function of astrocytes and play important roles in the homeostatic maintenance of extracellular potassium and glutamate levels in tripartite synapses. They also modulate the astrocytic expression of BDNF, a key molecule for neural plasticity and neurogenesis. The functions of Kir4.1 channels are attenuated in epileptic disorders and other brain diseases, including Huntington’s disease and neuropathic pain. In contrast, Kir4.1 channel functions are probably enhanced in major depressive disorder and autism spectrum disorder. Although further investigations are necessary, recent evidence suggest that Kir4.1 can serve as a glia-oriented novel target for the treatment of brain disease. In particular, Kir4.1 channel stimulators (e.g., channel activators or expressional enhancers) are anticipated to serve as therapeutic agents for the former diseases, Kir4.1 channel inhibitors (e.g., channel blockers or expressional suppressors) could be administered for the latter diseases.

**Table 1 ijms-22-10236-t001:** Pharmacological properties of Kir4.1 channel inhibitors.

Drugs	Kir4.1 Channel Inhibition		Binding Sites in Kir4.1	Pharmacological Actions	Ref.
	IC_50_ Value (Patch Configuration)	Voltage-Dependency			
**TCAs**					
Nortriptyline Desipramine Imipramine	38 μM (whole-cell)	Yes	E158, T128	Antidepressant action	[18]
**SSRIs**					
Sertraline Fluoxetine	7.2 μM 15.2 μM (whole-cell)	No	E158, T128	Antidepressant action	[19]
Quinacrine	1.8 μM (inside-out)	Yes	E158, T128	Antimalarial action	[21]
Chloroquine	ca. 0.5 μM (inside-out) ca. 7 μM (whole-cell)	Yes	E158, T128	Antimalarial action	[22]
Pentamidine	0.097 μM (inside-out)	Yes	E158, T127, T128	Antiprotozoal action	[23]
VU0134992	0.97 μM (whole-cell)	Yes	E158, Ile 159	Diuretic action	[24]
**Aminoglycosides**					
Gentamycin Neomycin Kanamycin	6.2 μM 63.8 μM 76.8 μM (inside-out)	Yes	E158, T128	Antibiotic action	[25]

**Table 2 ijms-22-10236-t002:** Pathophysiological changes in Kir4.1 channels in animal models and human disorders.

Kir4.1-Related Disorders	Changes in Kir4.1 Channels	Clinical Phenotype	Ref.
**Animal models**			
Kir4.1 conditional knockout mice	Kir4.1 channel dysfunction Reduced uptake of extracellular K^+^ and glutamate into astrocytes	Body tremor, Ataxia, High susceptibility to GTCSs, Premature death	[9,11]
Noda epileptic rat	Reduced expression of Kir4.1 in the amygdala (presynaptic processes)	Spontaneous GTCSs	[52]
*Lgi1* mutant rats	Reduced expression of Kir4.1 in the temporal lobe	High susceptibility to audiogenic kindling	[51]
Seizure susceptible DBA/2 mice	T262S mutation in Kir4.1 gene Dysfunction of Kir4.1 channel Reduced uptake of glutamate	High susceptibility to GTCSs,	[50]
Trauma-induced epilepsy rat	Chronic loss of Kir4.1 in the cerebral cortex (presynaptic processes)	Spontaneous partial seizures of cortical origin	[53]
**Human disorders**			
EAST/SeSaME syndrome	Loss-of-function mutations in Kir4.1 gene Kir4.1 channel dysfunction	GTCSs, Ataxia, Deafness, Mental retardation, Increased K^+^ excretion	[16,55,56,57]
Mesial temporal lobe epilepsy	Reduced Kir4.1 expression in the temporal lobe	Refractory partial seizures	[58]
Temporal lobe epilepsy with hippocampal sclerosis	Reduced Kir4.1 expression in the hippocampus	Refractory partial seizures	[59]
Focal cortical dysplasia type1	Reduced Kir4.1 expression in seizure foci	Refractory partial seizures	[60]
Glioma with epilepsy	Reduced Kir4.1 expression in the glioma site	GTCSs	[61]
Lethal epileptic disorder	Double mutations in Kir4.1 and SLACK genes Kir4.1 channel dysfunction	Severe-disabling seizures Developmental delay Early death	[62]

SLACK: Na^+^-activated K^+^ channel subunit.

## Data Availability

Not applicable.

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
