# Peer review of "Emerging Roles of Astrocyte Kir4.1 Channels in the Pathogenesis and Treatment of Brain Diseases"

_ijms, 2021, doi:10.3390/ijms221910236_

Round 1
Reviewer 1 Report
In this manuscript, the authors summarize and discuss about the roles of astrocyte Kir4.1 channels in the pathogenesis and treatment of brain diseases such as major depressive disorder and epilepsy. Firstly this review introduces thoroughly the physiological functions and fundamental roles of Kir4.1 in tripartite synapses, especially focusing on the spatial K+ buffering and glutamate signaling. Then additionally, the authors summarize the pathophysiological implications of Kir4.1 channels in CNS diseases and also highlight the therapeutic potential of Kir4.1 channel modulators. This review seems to cover the recent findings greatly, and thus it will be of great interest to the broad readership of this journal. This reviewer does not have any additional requests.
Author Response
We thank the Reviewer very much for his/her careful review and evaluation of our manuscript, appreciating positive and constructive comments.
Reviewer 2 Report
Ohno et al reviewed the role and pharmacological features of astrocytic Kir4.1 potassium channel in selected neurological diseases, such as major depressive disorder, epilepsy, and to the lesser extent in Huntington’s disease, Autism spectrum disorder, and neuropathic pain. Indeed, there is increasing evidence implicating altered potassium buffering in different neurological pathologies. While the topic is interesting, the information presented in this manuscript does not fully help readers understand the rationale and significance of this line of the review.
The role of Kir4.1 in astrocyte and brain metabolism is clearly a critical factor in multiple interactive pathways and is an anchoring point for neurotransmission. This topic of the review is not unique and several reviews are available. The question here is: how is the insight of this review different and how does it add to the field. The unique insights and potentially new concept(s) that this review might propose should be clearly defined.
There is a general agreement among researchers that impaired Kir4.1-mediated K+ homeostasis may result in seizures, brain edema, and decreased neurotransmitter glutamate reuptake, leading to excitotoxicity, a fact rather documented coincidentally Kir4.1 involvement in different pathologies.
The authors claim support for the idea that the astrocytic Kir4.1 channel could potentially serve as a novel therapeutic target for the treatment of depressive disorders and epilepsy. However, the scenario is more complex since the Kir4.1 does not act alone and the Kir4.1 changes are mostly the effect of ion imbalance. Some background information about the astrocytic unique function, especially in the context of potassium buffering is needed. What about other channels/transporters/processes that control K+ buffering in astrocytes? Since clear-cut evidence is lacking it is difficult for readers to understand the meaning of this concept without the more complex picture of astrocytic ion homeostasis. What about transcriptional/posttranslational/epigenetic control of Kir4.1 expression? Does available data serve a clear picture?
Other comments:
Potentially new and interesting data reefer pharmacological Kir4.1 regulation. However, more detailed information would be of value e.g. type of inhibition, EC(50), type of action, etc. additionally, a table with pharmacological features would be of value. A defined statement and conclusion delivered from pharmacology studies will be welcome.
Kir4.1 regulating role in the astrocytic expression of BDNF in tripartite synapses is highly premature and unclear. MDD is often related to BDNF gene mutations, how this fact can be reconciled with this proposed by authors?
p.1 line 12-13 Kir4.1 function (one) cannot be summarized,
p.5 line 161; how Kir4.1 may induce brain diseases; this is mostly the effect of observed pathological changes;
What about the known redistribution of Kir4.1 channels in epilepsy? Does it contribute to this pathology?
Table summarizing clinical and pre-clinical data would be of value.
The limitations of the review concept should be defined. Since, the authors partially defined questions/issues that remain to be solved, the way how might this publication impact basic science and clinical practice, should be specified.
Author Response
We thank the Reviewer very much for his/her invaluable and constructive comments, which improve the readability of our article. According to the Reviewer's comments, we revised our manuscript.
To overall comments
We agree with the Reviewer that some of the features and roles of Kir4.1 channel in modifying brain diseases are still immature. Since astrocytes have tremendous functions for which numerous molecules and factors are interacting each other in complex ways, we agree that this review have some limitations for understanding. We also do not deny any possible involvement of other factors or molecules (e.g., channels/transporters/processes) in controlling K+ buffering in astrocytes.
In this article, we reviewed pharmacological features of Kir4.1 channels and discussed their involvement in the neurobiological and pathophysiological basis of brain diseases, highlighting the therapeutic potential of Kir4.1 channel modulators. Even with some limitations, we believe, this review provides important information on pathophysiological role of astrocyte Kir4.1 channels in modifying brain disease conditions and on drug discovery (medicinal chemistry) for identifying novel Kir4.1 ligands. These are important to facilitate the paradigm shift from classical (neuron-related) medicines to creating novel glia-based medications. Meanwhile, in response to Reviewer’s comments, we stated limitations of this review and necessary future investigations in the revised manuscript (p.11 line 386-401).
To other comments:
Potentially new and interesting data reefer pharmacological Kir4.1 regulation. However, more detailed information would be of value e.g. type of inhibition, EC(50), type of action, etc. additionally, a table with pharmacological features would be of value. A defined statement and conclusion delivered from pharmacology studies will be welcome.
Thank you. In response to Reviewer’s comments, we added a Table (p. 5 Table 1), which summarizes the pharmacological features of Kir4.1 inhibitors, and also revised the text (p.4 line 140-142, p.5 line 153-158)
Kir4.1 regulating role in the astrocytic expression of BDNF in tripartite synapses is highly premature and unclear. MDD is often related to BDNF gene mutations, how this fact can be reconciled with this proposed by authors?
We agree that role and mechanisms of Kir4.1 channels in regulating BDNF expression remain unclear and made the statement for this and necessity of further studies, particularly in vivo studies using animal models and human patients, in the last paragraph of the manuscript (p.11 line 397-401).
p.1 line 12-13 Kir4.1 function (one) cannot be summarized,
Revised (p.1 line 13-14).
p.5 line 161; how Kir4.1 may induce brain diseases; this is mostly the effect of observed pathological changes;
Revised (p.6 line 175-176).
What about the known redistribution of Kir4.1 channels in epilepsy? Does it contribute to this pathology?
Thank you. Little is known about redistribution of Kir4.1 channels so far. But, some studies show abnormal trafficking (distribution) of Kir4.1 in animal models of Epilepsy. In response to Reviewer’ s comment, we added this information in the text (p.9 line 288-290).
Table summarizing clinical and pre-clinical data would be of value.
In response to Reviewer’s comments, we added a Table (p.8 Table 2) which summarizes pre-clinical and clinical data on Kir4.1 alterations in epileptic disorders.
The limitations of the review concept should be defined. Since, the authors partially defined questions/issues that remain to be solved, the way how might this publication impact basic science and clinical practice, should be specified.
As previously replied, we summarized pharmacological features of Kir4.1 channels and discussed their involvement in the neurobiological and pathophysiological basis of brain diseases. Specifically, we highlighted the therapeutic potential of Kir4.1 channel stimulators (e.g., channel activators and expressional enhancers) for the treatment of epileptic disorders, Huntington’s disease and neuropathic pain, and of Kir4.1 channel inhibitors (e.g., channel blockers and expressional suppressors) for depressive disorders and autism, as shown in Figure 4. Even with limitations, we believe that the review provides information on pathophysiological role for astrocyte Kir4.1 channels in modifying brain disease conditions and on drug discovery research for identifying novel Kir4.1 ligands, facilitating a paradigm shift from classical (neuron-related) to novel glia medications. We state these in the “Concluding Remarks” section.
In addition, according to Reviewer’s comments, we also add the statement on limitations of the review and necessary future investigations (p.11 line 386-401).
Reviewer 3 Report
In this manuscript (“Emerging roles of astrocyte Kir4.1 channels in the pathogenesis 2 and treatment of brain diseases”) the aim of the authors was to highlight the therapeutic potential of the inwardly rectifying potassium Kir4.1 channel modulators in brain diseases, specifically Kir4.1 channel stimulators and Kir4.12 channel inhibitors. In this review, the authors summarize the functions of Kir4.1 channels in astrocytes and discuss their therapeutic potential. The authors have collected the publications dealing with the role of Kir4.1 channels in brain diseases: major depressive disorder, epileptic disorders. Furthermore, they claim that Kir4.1 channels are involved in a number of brain diseases, including Huntington’s disease, autism spectrum disorders, Parkinson’s disease, Alzheimer’s disease, amyotrophic lateral sclerosis, neuropathic pain. Astrocytes are the most abundant glial cells and are involved in maintaining the integrity of brain functions.
The role of astrocytes in potassium (K+) buffering is of particular importance. Kir4.1 channel-mediated spatial K+ buffering is also important for glutamate and water homeostasis in synapses, as Kir4.1 channels are functionally coupled to the excitatory amino acid transporters (EAATs).
It is known that astrocytes are tightly interconnected, form blood-brain barriers with microcapillaries. These cells are essential for maintaining normal brain functions and are thought to be closely involved in the pathogenesis and treatment of brain diseases.
Kir4.1 channels and EAATs in astrocytes have been shown to be reduced in patients ? in number with Huntington’s disease and animal models.
Neuropathic Kir4.1 channels are down-regulated in animal models of neuropathic pain. Several compounds including antidepressant drugs (e.g., fluoxetine) bind to and inhibit Kir4.1 channel-mediated currents.
As far as the role of Kir4.1 channels in regulating the extracellular levels of K+ and glutamate, and the 362 astrocytic expressions of BDNF in tripartite synapses they concluded that synthetic molecules are able to influence Kir4.12 channels are potential candidates for drug development.
Concern:
The problem of dietary K+ intake (plays an important role in the regulation of Kif4.1/Kir5.1) should be discussed (Xiao-Tong Su,1 David H. Ellison,2,3 and Wen-Hui Wang Kir4.1/Kir5.1 in the DCT plays a role in the regulation of renal K+ excretion Am J Physiol Renal Physiol. 2019 Mar 1; 316(3): F582–F586. Published online 2019 Jan 9. doi: 10.1152/ajprenal.00412.2018 PMCID: PMC6459306PMID: 30623727
Fluoxetine (antidepressant) has got another important effect ( Vizi, E. S., Kisfali, M., Lőrincz, T., Role of nonsynaptic GluN2B-containing NMDA receptors in excitotoxicity: evidence that fluoxetine selectively inhibits these receptors and may have neuroprotective effects. Brain Res. Bull. 93: 32-8 (2013)). Worth mentioning when you discuss depression (4.1 Major depressive disorders): “The relative potencies of antidepressants for Kir4.1 channel inhibition (sertraline > fluoxetine > imipramine >> fluvoxamine > mianserin) are consistent with those for BDNF induction.”.
The manuscript needs some correction of misspelt words.
Author Response
We thank the Reviewer very much for his/her invaluable and constructive comments, which improve the readability of our article. According to the Reviewer's comments, we revised our manuscript.
The problem of dietary K+ intake (plays an important role in the regulation of Kif4.1/Kir5.1) should be discussed (Xiao-Tong Su,1 David H. Ellison,2,3 and Wen-Hui Wang Kir4.1/Kir5.1 in the DCT plays a role in the regulation of renal K+ excretion Am J Physiol Renal Physiol. 2019 Mar 1; 316(3): F582–F586. Published online 2019 Jan 9. doi: 10.1152/ajprenal.00412.2018 PMCID: PMC6459306PMID: 30623727
We cited the reference suggested by Reviewer and revised the manuscript (p. 11 line 392-394).
Fluoxetine (antidepressant) has got another important effect ( Vizi, E. S., Kisfali, M., Lőrincz, T., Role of nonsynaptic GluN2B-containing NMDA receptors in excitotoxicity: evidence that fluoxetine selectively inhibits these receptors and may have neuroprotective effects. Brain Res. Bull. 93: 32-8 (2013)). Worth mentioning when you discuss depression (4.1 Major depressive disorders): “The relative potencies of antidepressants for Kir4.1 channel inhibition (sertraline > fluoxetine > imipramine >> fluvoxamine > mianserin) are consistent with those for BDNF induction.”.
We cited the reference suggested by Reviewer and revised the manuscript (p.6 line 165-166).
Round 2
Reviewer 2 Report
The Authors sufficiently addressed all the issues. One minor suggestion is the lack of references in the newly added Tables 1 and 2 that can be easily inserted. Although all references are specified in the main text, this format would be more convenient for the readers.
Author Response
Thank you for your comments.
In response to Reviewer's comments, we revised Tables with putting References #. We add 2 references and revised text.
All changes are colored as green